# Changes and Associations between Cervical Range of Motion, Pain, Temporomandibular Joint Range of Motion and Quality of Life in Individuals with Migraine Applying Physiotherapy: A Pilot Study

**DOI:** 10.3390/medicina57060630

**Published:** 2021-06-17

**Authors:** Egle Lendraitiene, Laura Smilgiene, Daiva Petruseviciene, Raimondas Savickas

**Affiliations:** Department of Rehabilitation, Medical Academy, Lithuanian University of Health Sciences, Eiveniu str. 2, LT-50161 Kaunas, Lithuania; egle.lendraitiene@lsmuni.lt (E.L.); laura.smilgiene@gmail.com (L.S.); raimondas.savickas@lsmuni.lt (R.S.)

**Keywords:** migraine physiotherapy, exercise, migraine quality of life, pressure pain thresholds, cervical range of motion, temporomandibular joint and migraine

## Abstract

*Background and Objectives*: The aim of this study was to assess the effects of physiotherapy with aerobic exercise together with temporomandibular joint range of motion exercises (supervised) and physiotherapy with aerobic exercise only (unsupervised), also to review the correlations between neck movements, pain, temporomandibular joint range of motion movements and quality of life in individuals with migraine. *Methods*: The flexion, extension and lateral flexion of the cervical spine were measured in degrees with a mechanical goniometer and pressure pain thresholds with algometer. Quality of life was assessed with the SF-36 questionnaire and temporomandibular joint range of motion with a centimeter. *Results*: The study showed statistically significant cervical flexion results in both groups (*p* < 0.05), masticatory muscle results and temporomandibular joint range of motion between the groups (*p* < 0.05). A correlation between left upper trapezius muscle pain and cervical lateral flexion was observed in the intervention group. Physical activity correlated with cervical extension, activity limitation due to physical ailments and general health. A correlation between temporomandibular joint and right-side masticatory muscles pain was found. A correlation between upper trapezius muscle pain and left- as well as right-side temporalis muscles were found in the control group. Strong correlations were found between pain and activity limitation due to physical ailments and emotional state. The temporomandibular joint range of motion strongly correlated with activity limitation due to physical ailments. *Conclusions:* Physiotherapy based on aerobic exercises together with temporomandibular joint exercises was more effective than physiotherapy based on aerobic exercise for decreasing pain, increasing pressure pain thresholds and cervical range of motion.

## 1. Introduction

Migraine is a very serious health problem and affects not only the individual, but also society. It is associated with a high economic burden and a significant deterioration in the patient quality of life [1]. It is one of the most common nervous system disorders causing the greatest disability in individuals aged 15 to 49 years worldwide. A total of 15 to 18 percent of the adult population is afflicted with migraine [2,3]. More than 90 percent of individuals have reported a headache at least once in their lifetime and every second adult has had one it at least once in the past year [3].

Epidemiological data have revealed that two-thirds of migraine patients have anxiety disorders and migraineurs with functional somatic disorders (e.g., fibromyalgia) are more likely to experience mood swings [4]. Recently, more attention has been paid to pharmacological treatment; however, some patients have poor tolerance for it due to existing side effects, and the efficacy may also be insufficient [5].

Therefore, non-pharmacological methods such as physiotherapy, acupuncture and various relaxation techniques are applied for individuals with migraine [6]. Sleep disorders and circadian dysregulation appear to be associated with migraine, so melatonin may be used in the treatment and prevention of migraine. However, the complementary RCTs evidence is required [7]. Furthermore, minerals and vitamins may be associated with migraine. Vitamin supplementation, especially vitamin B2 or riboflavin, is used as an effective adjunct therapy. Studies reveal that it is a safe, well-tolerated and inexpensive measure of preventing migraines in adults. Vitamin D has already been associated with several neurological disorders and several cases reported vitamin D benefit to musculoskeletal parameters, similar to physiotherapy [8]. The beneficial effect of B6, B12, and folate in the prophylaxis of migraine treatment has been indicated [9]. CoQ10 also has beneficial effects in reducing the frequency and duration of migraine attacks [10]. So, lifestyle, dietary or behavioral interventions can be involved in the treatment and prevention of migraine [8].

Many different health organizations recommend regular physical activity and exercise for the treatment and prevention of migraine. However, the relationship between aerobic exercise and migraine is still not fully understood: the optimal intensity or whether the benefits of aerobic exercise are synergistic with pharmacological treatment, the mechanisms of which validate its effectiveness as a treatment [11].

Primary headache has also been reported to be associated with neck or shoulder girdle muscle pain, which is most commonly expressed as myofascial trigger points [12]. Ferracini et al. reported that individuals with migraine suffer from active trigger points, but it is not entirely clear how it affects the frequency and intensity of migraine attacks [13]. Other authors found that active trigger points may lead to headache, temporomandibular joint disorder or migraine [14].

## 2. Methods

### 2.1. Design

The study was a “parallel group” randomized controlled clinical trial evaluating six weeks of physiotherapy with aerobic exercise.

### 2.2. Ethics

The local ethical committee of the Lithuanian University of Health Sciences approved the study (protocol no. BEC-SR(M)-266). ClinicalTrials.gov Identifier: NCT04729699. All the patients gave written informed consent.

### 2.3. Participants

Participants took prophylactic medications such as vitamin supplements during the trial. The inclusion criteria consisted of: (1) diagnosis of migraine by a neurologist based on criteria of the International Classification Headache Disorders-III (ICHD-3); (2) aged between 18 and 50 years; (3) patients fulfilling criteria for more than one primary headache disorder were included.

Participants were excluded if they had chronic migraine, medication overuse headache and presented any of following conditions: (1) any other neurological or psychiatric disorders, (2) head or neck injuries, cervical spine hernia, (3) face and temporomandibular injuries, (4) orthodontic problems or persons using muscle relaxants.

### 2.4. Procedure

All the participants were women. They were randomized into two groups using a randomly generated sequence by a computer (Table 1). The assessor was blinded to the subject’s group assignments.

Cervical range of motion was assessed with mechanical goniometer. Temporomandibular joint (TMJ) range of motion was measured with a centimeter strip. Pressure pain thresholds were measured with an algometer and quality of life was assessed with an SF-36 questionnaire at baseline and at the end of treatment.

### 2.5. Primary Outcomes

Movements were measured in degrees with a mechanical goniometer in a standardized sitting position. The flexion and extension of the cervical spine were measured in the sagittal plane about the transverse axis and lateral flexion of the cervical spine—in the frontal plane about the lateral axis [15].

#### SF-36 Questionnaire

Quality of life was assessed with the SF-36 questionnaire. It consists of 36 questions from eight areas of life: pain, general health, energy, physical activity, activity limitation due to emotional disorders or physical problems, social connections and psycho-emotional state. Physical health was assessed by physical activity, activity limitation due to physical problems or pain; general health and mental health was reflected by activity limitation due to emotional state, social connections and energy. Answers were scored (total score 100). A higher total score meant a better quality of life. The questionnaire was completed at baseline at the end of treatment [16].

Pressure pain thresholds (PPT). The Wagner FPX electronic digital algometer was used to measure the pain threshold in the temporal, upper trapezius and masticatory muscles. Each muscle point was measured three times and only the average was recorded. Pressure values ranged from 0 to 10; measurements are expressed in kg/cm^2^ [17].

### 2.6. Secondary Outcomes

#### 2.6.1. Anthropometry

TMJ range of motion was measured with a centimeter before and after the physiotherapy. Range of motion was measured three times and only the average was recorded.

#### 2.6.2. Exercise Group

The exercise group, supervised by a physiotherapist, was performed outdoors. Twelve sessions of physiotherapy with aerobic exercises were performed for 30 to 45 min each.

The physiotherapy program followed two sessions per week for six weeks. Each session included a 10-min warm up (static breathing exercises and light stretching exercises were performed). The main part consisted of 10 min of fast walking outside, 5 min of light running, as well stretching exercises and strength exercises for upper (wall pushups, T- pushups, press ups) and lower limbs (supine leg raise, squats, bridge, forward/side steps up) were included. Cooldown involved breathing and relaxation exercises. Additionally, TMJ exercises were performed. This part consisted of masticatory and temporalis muscles massage, mouth opening movement, mouth opening in alignment and sideways movement.

#### 2.6.3. Control Group

The control group was introduced to the same physiotherapy program as the exercise group (only without TMJ exercises) and was trained to perform it independently, unsupervised, at home.

### 2.7. Statistical Analysis

Statistical analysis was performed using IBM SPSS Statistics 22 software and MS Excel 2016 program. Quantitative survey data are presented as the median (x_me_), minimum value (x_min_), maximum value (x_max_) and mean (x) − (x_me_ (x_min_–x_max_; $\hat{x}$)). The Wilcoxon test was used to compare two dependent samples and the Mann–Whitney test for two independent samples. Statistical significance was assumed when *p* < 0.05. The Spearmen correlation coefficient (r) was calculated to determine the correlation. The relationship when 0.00 < |r| ≤ 0.39 was considered very weak, 0.4 < |r| ≤ 0.69 was moderate, 0.7 < |r| ≤ 0.89 was considered strong and 0.9 < |r| ≥ 1 was very strong [18].

## 3. Results

Statistically significant differences were found in cervical lateral flexion to the right (*p* = 0.035) and TMJ range of motion (*p* = 0.022) between groups (Table 2). Statistically significant masticatory left-side muscle results were noted between groups (*p* = 0.035) (Table 3).

Quality of life was assessed with an SF-36 questionnaire. Only emotional state results were significantly increased between groups (*p* = 0.035) (Table 4).

The correlation between left upper trapezius muscle pain and cervical lateral flexion was observed in the exercise group (r = 0.816, *p* = 0.004). Physical activity strongly correlated with cervical extension (r = 0.719, *p* = 0.019), activity limitation due to physical ailments (SF-36) (r = 0.763, *p* = 0.010), and general health (SF-36) (r = 0.880, *p* = 0.001). A correlation between TMJ and right-side masticatory muscles pain was found (r = 0.781; *p* = 0.013).

A correlation between upper trapezius muscle pain and left- (r = 0.813; *p* = 0.010) as well as right-side temporalis muscles (r = 0.811; *p* = 0.008) were found in the control group. Strong correlations were found between pain and activity limitation due to physical ailments (SF-36) (r = 0.786; *p* = 0.012) and emotional state (SF-36) (r = 0.740; *p* = 0.014). TMJ range of motion also strongly correlated with activity limitation due to physical ailments (SF-36) (r = 0.824; *p* = 0006) (Table 5).

## 4. Discussion

Migraine is the second most common primary headache disorder and the third most common disease in the world (behind dental caries and tension-type headache), causing disability in individuals under 50 years of age.

Unfortunately, it is still insufficiently diagnosed and treated [19]. In this study, we evaluated the effect of physical therapy with aerobic exercise on cervical range of motion, pain, TMJ range of motion and quality of life, and compared the results to assess the associations between variables.

Earlier studies [20,21] have investigated the effect of physiotherapy for individuals with migraine. Methods differed from ours and it is difficult to compare the results. Individuals with migraine often complain about neck pain and it usually worsens the headaches. Individuals with neck pain usually have reduced cervical range of motion and studies assessing it are scarce. However, in one study, cervical range of motion was significantly improved compared to the control group and no differences were found in groups and between the groups in another study.

We found a statistically significant improvement in cervical range of motion (flexion, extension, cervical flexion) in exercise group and only cervical flexion was improved in the control group. A study by Grossi et al. showed significant improvement in cervical range of motion in an exercise group compared with control [20].

Szikszay et al., in their systematic review and meta-analysis, found that physiotherapy significantly improved PPT in temporalis and masticatory muscles, but no changes were observed in the trapezius [22]. Another study showed reduced pain and increased PPT for masticatory and temporalis muscles applying physiotherapy and manual therapy [23]. Additionally, Andersen et al., in their systematic review, found that women reported lower PPT in temporalis and masticatory muscles compared with men with migraine [24]. Our study showed improved PPT in temporalis and upper trapezius in exercise groups, and only PPT in masticatory muscles were significant improved in both groups and between the groups.

Pedron et al., in a 12-week study, found that physical therapy with TMJ exercises improved TMJ range of motion compared with the group that received physiotherapy alone. Only pain was decreased in both groups [25]. Other authors have investigated interventions based on manual therapy and physiotherapy for TMJ. They found that it better improves TMJ range of motion compared to splints. However, studies are limited, and the results should be assessed cautiously [26]. Our results overlapped with other authors’ results and improved TMJ range of motion was demonstrated in the exercise group and also between the groups.

In our study, quality of life was assessed. Physiotherapy improved general health, physical activity, and activity limitation due to physical ailments in both groups. Decreased pain and increased energy was reported in the exercise group.

The results of meta-analysis [27] showed that aerobic exercise reduced pain, migraine frequency and duration as well as improving the quality of life in individuals with migraine. Varkey et al.’s 12-week study revealed that patients who performed indoor cycling three times per week decreased the instance of headache or migraine attacks and reported an improved quality of life [28]. Another study evaluated 88 patients’ quality of life with an SF-36 questionnaire in which all aspects, including physical activity, activity limitation due to physical ailments, pain, general health and others, were improved in the exercise group compared to the control group (*p* ˂ 0.05) [29].

Correlations were found between the upper trapezius and cervical lateral flexion in the exercise group in our study. The emotional state correlated with pain and physical activity correlated with cervical extension, activity limitation, and general health. Additionally, a correlation between TMJ and right-side masticatory muscles was found. In the control group, a relationship was found between the upper trapezius and right-side temporalis muscle. Pain correlated with activity limitation due to physical ailments and emotional state. TMJ range of motion strongly correlated with activity limitation due to physical ailments.

One study revealed that migraine attacks are related to trigger points in the masticatory muscles, which cause myofascial pain [30]. Calixtre et al. identified that pain and headache were strongly associated with general health in individuals with migraine with TMJ disorders [31]. Another study identified that TMJ disorders were more common in individuals with migraine [32].

Researchers have also found that pain was associated with depression and deteriorated quality of life. However, another study did not find any correlations between pressure pain in TMJ and upper trapezius muscles [33].

Aerobic exercise benefits the musculoskeletal system but may provide additional benefits for patients with migraine because it significantly inhibits inflammatory cytokines. Togha et al. underlined that with increasing levels of inflammatory factors, headaches tended to be more chronic. However, to confirm the hypothesis that neuroinflammation plays a role in migraine pain genesis, experimental and randomized controlled trials are required [34]. Liampas et al. noted that lipid abnormalities may play an important role in the increased vascular risk of migraine patients, but preventive medications and aerobic exercises may reduce their concentration [35]. Moreover, it helps to reduce oxidative and metabolic stress, which is associated with increased homocysteine in plasma. Homocysteine catalyzes vitamins B6, B12, and folic acid, which can reduce the severity of migraines with aura. Therefore, these vitamins may be useful prophylactic drugs for the treatment of migraines with aura [36].

Several limitations of this study should be acknowledged. First, the small number of participants does not allow for generalization. However, future studies should be carried out including more participants. Second, most of the participants did not stop pharmacological treatment (preventive treatment) during the study and any increase/decrease in drug intake was not analyzed. Third, migraine frequency, intensity, and duration were unreported, and should be analyzed in future studies.

## 5. Conclusions and Recommendations

Physiotherapy with aerobic exercises performed with TMJ exercises were more effective than physiotherapy alone for decreasing pain, increasing PPTs and cervical range of motion.

The results of our study and the other authors’ research suggest that physiotherapy could be an effective tool in migraine treatment. We assume that supervised physiotherapy based on aerobic exercise could be an effective alternative and non-invasive method in the treatment of migraine, but further studies are required. As the effect of physiotherapy was a significant event in the control group, further investigations are encouraged.

## Figures and Tables

**Table 1 medicina-57-00630-t001:** Clinical characteristics of persons with migraine.

	Exercise (*n* = 10)	Control (*n* = 9)
Gender, *n*		
Female	10	9
Age (years)	48.5 (37–50; 45.2)	47 (42–49; 46.1)
Migraine duration (years)	12 (7–20; 12.7)	12 (8–17; 12.2)
Type of migraine:		
Migraine with aura	6 (50%)	5 (62%)
Migraine without aura	4 (40%)	4 (38%)

Numbers are presented as median, minimum, maximum, mean (x_me_ (x_min_–x_max_; $\hat{x}$)) or number (%).

**Table 2 medicina-57-00630-t002:** Results of cervical range of motion in groups.

	Exercise (*n* = 10)		Control (*n* = 9)		Exercise	Control	between Groups
	Baseline	End	Baseline	End	*p*-Value ^a^ (95% CI)		
Cervical flexion (°)	34.5 (28–41; 34.5)	37.5 (30–42; 36.4)	34 (28–40; 33.4)	35 (30–41; 34.9)	0.007 ^b^	0.026 ^b^	0.497
Cervical extension (°)	29 (27–31; 28.9)	30 (28–32; 29.7)	28 (26–30; 28)	28 (27–30; 28.4)	0.011 ^b^	0.059	0.053
Lateral flexion to right (°)	38 (34–39; 37.2)	38.5 (35–40; 37.9)	35 (33–38.5; 35.4)	35 (33–38.9; 35.8)	0.011 ^b^	0.066	0.035 ^b^
Lateral flexion to left (°)	35 (33–39; 35.6)	36 (34–39; 36.3)	35 (34–37; 35.3)	35.2 (34–37; 35.4)	0.020 ^b^	0.157	0.497
TMJ range of motion (cm)	4.4 (4–5.4; 4.54)	4.7 (4.15–5.8; 4.85)	3.8 (3.6–5.8; 4.24)	3.85 (3.65–5.3; 4.21)	0.005 ^b^	0.206	0.022 ^b^

^a^*p*-value < 0.05. ^b^ Significant result.

**Table 3 medicina-57-00630-t003:** Results of pressure pain thresholds (PPT) in groups.

Pressure Pain Tresholds (PPT)	Exercise (*n* = 10)		Control (*n* = 9)		Exercise	Control	between Groups
	Baseline	End	Baseline	End	*p*-Value ^a^ (95% CI)		
M. temporalis right side	1.48 (1.23–1.58; 1.46)	1.56 (1.31–1.69; 1.53)	1.56 (1.49–1.62; 1.55)	1.56 (1.50–1.62; 1.55)	0.005 ^b^	0.102	1.000
M. temporalis left side	1.55 (1.41–1.66; 1.54)	1.62 (1.41–1.71; 1.59)	1.49 (1.39–1.64; 1.52)	1.50 (1.39–1.64; 1.52)	0.008 ^b^	0.059	0.113
M. upper trapezius right side	1.26 (0.76–1.68; 1.26)	1.36 (0.94–1.79; 1.39)	1.12 (0.76–1.52; 1.15)	1.12 (0.77–1.52; 1.15)	0.005 ^b^	0.083	0.113
M. upper trapezius left side	1.28 (0.89–1.47; 1.22)	1.32 (0.99–1.49; 1.28)	1.13 (0.98–1.61; 1.21)	1.13 (0.99–1.61; 1.22)	0.005 ^b^	0.059	0.497
M. masticatory right side	2.07 (1.68–2.78; 2.16)	2.16 (1.71–2.86; 2.24)	2.02 (1.69–2.32; 1.96)	2.07 (1.73–2.41; 2.02)	0.005 ^b^	0.008 ^b^	0.315
M. masticatory left side	2.21 (1.74–2.79; 2.25)	2.33 (1.86–2.84; 2.34)	2.01 (1.75–2.10; 1.94)	2.07 (1.84–2.17; 2.01)	0.005 ^b^	0.007 ^b^	0.035 ^b^

^a^*p*-value < 0.05. ^b^ Significant result.

**Table 4 medicina-57-00630-t004:** Results of quality of life in individuals with migraine in groups.

SF-36 (Score 0–100)	Exercise (*n* = 10)		Control (*n* = 9)		Exercise	Control	between Groups
	Baseline	End	Baseline	End	*p*-Value ^a^ (95% CI)		
Physical activity	64 (61; 77–65.7)	69 (62–77; 69.1)	65 (61–72; 66.1)	67 (62–72; 67.3)	0.008 ^b^	0.016 ^b^	0.447
Activity limitation due to physical ailments	62.5 (50–75; 62.5)	75 (50–100; 80)	50 (50–75; 61.1)	75 (50–100; 72.2)	0.008 ^b^	0.046 ^b^	0.447
Pain intensity	77.85 (67–100; 81.1)	99.8 (78–100; 91.1)	72.7 (67–100; 80.2)	78 (67–100; 84)	0.028 ^b^	0.068	0.182
General health	65 (55–75; 64.5)	72.5 (60–85; 72)	70 (60–75; 67.8)	75 (65–80; 74.4)	0.007 ^b^	0.006 ^b^	0.549
Energy	70 (60–80; 70)	75 (65–88; 74.5)	70 (60–80; 71.1)	75 (65–80; 72.3)	0.003 ^b^	0.083	0.604
Social function	67 (67–78; 71.4)	72.5 (67–88; 74.4)	67 (56–78; 68.2)	67 (67–78; 70.1)	0.102	0.157	0.400
Activity limitation due to emotional disorders	67 (67–100; 80.2)	100 (67–100; 90.2)	67 (33–100; 63)	67 (67–100; 78)	0.063	0.059	0.156
Emotional state	72 (68–80; 73.6)	75.8 (68–80; 75.1)	72 (64–76; 68.9)	72 (64–76.3; 70.3)	0.063	0.068	0.035 ^b^

^a^*p*-value < 0.05. ^b^ Significant result.

**Table 5 medicina-57-00630-t005:** Associations between cervical range of motion, pain, temporomandibular joint range of motion and quality of life in individuals with migraine.

Associations	Pain of Left M. Upper Trapezius	Activity Limitation Due to Physical Ailments (SF-36)	General Health (SF-36)	Pain of Right Masticatory Muscles
Exercise group				
Lateral flexion to left	r = 0.816, *p* = 0.004 **			
Cervical extension		r = 0.719, *p* = 0.019 **		
TMJ range of motion				r = 0.781, *p* = 0.013 **
Physical activity (SF-36)		r = 0.763, *p* = 0.010 **	r = 0.880, *p* = 0.001 **	
	Pain of left M. temporalis	Pain of right M. temporalis	Emotional state (SF-36)	Activity limitation due to physical ailments (SF-36)
Control group				
Pain of upper trapezius right side	r = 0.813, *p* = 0.010 **	r = 0.811, *p* = 0.008 **		
Pain intensity (SF-36)			r = 0.740, *p* = 0.014 **	r = 0.786, *p* = 0.012 **
TMJ range of motion				r = 0.824, *p* = 0.006 **

**—strong correlation.

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
