# Peer review of "Changes and Associations between Cervical Range of Motion, Pain, Temporomandibular Joint Range of Motion and Quality of Life in Individuals with Migraine Applying Physiotherapy: A Pilot Study"

_medicina, 2021, doi:10.3390/medicina57060630_

Round 1

Reviewer 1 Report

Authors performed a randomized controlled trial to assess the effect of physical therapy plus exercise on migraine. Authors found beneficial effects in both the experimental and the control group, while the benefits of the active program over control were only slightly significant.

I have several concerns over this manuscript.

1 - The terms "physical therapy" and "exercise" are used interchangeably in this paper, while they refer to different kinds of interventions. The literature cited by the Authors refers to both exercise and physical therapy. Besides, the interventional program of the present paper included both exercise (walking, running, strength exercises) and physical therapy (breathing, stretching). In my understanding, the exercise component was more important than physical therapy in this study.

2 - I suggest describing the interventions in a more detailed fashion. In my understanding, the active group was assisted in the intervention by professionals, while the control group essentially followed the same protocol on their own. The lack of difference between groups was probably due to a high compliance to the control intervention in the control group, leading to significant results despite the absence of direct supervision by professionals. Please report and discuss.

3 - The lack of significant differences between the active and control group might come from low numbers leading to low statistical power. How did Authors calculate their sample size?

4 - No outcome is reported referring to migraine status, including monthly migraine days and acute drug consumption. If those outcomes were not considered, which was the rationale of focusing a study on patients with migraine?

5 - Was any patient on treatment with migraine preventive drugs?

6 - In the Results, Authors mention "associations", while in fact they report measures of correlation.

7 - In the Discussion, a thorough report on the study limitations is lacking.

8 - The Conclusions are little more than a mere repetition of the Results. I suggest focusing on the interpretation of the study in light of current literature.

Author Response

Thank you very much for your questions and help.

Best regards, 

Daiva Petruseviciene

Reviewer 2 Report

Extensive English editing is required. Please address this issue in your revision. Also, please replace the term physiotherapy with aerobic exercise throughout the text (it is a more accurate term)

Title

Please state that your study is an RCT (A randomized controlled pilot study)

Introduction

  1. When you mention that migraine ‘‘is one of the most common nervous system disorders in individuals aged 15-49 years worldwide causing the greatest disability’’, please specify the greatest disability compared to which disorders (among nervous system disorders? among all disorders?) and patients (all individuals? 15-49 years old individuals?)
  2. The last second of the 1st Paragraph is redundant, please remove it. Headache in general is not of interest. Please focus on migraine.
  3. Please avoid the word ‘‘migraineurs’’ and replace with patients/individuals with migraine.
  4. In the context of the non-pharmacological methods, you only refer to physiotherapy, acupuncture and relaxation techniques. Please expand this part by commenting on the increasingly popular use of alternative preventive medicines for migraine, and most notably the following neutraceuticals: melatonin (please cite: Endogenous Melatonin Levels and Therapeutic Use of Exogenous Melatonin in Migraine: Systematic Review and Meta-Analysis. Headache, 2020), vitamin D (please cite: Vitamin D serum levels in patients with migraine: A meta-analysis. Rev Neurol, 2020), riboflavin (please cite: Prophylaxis of migraine headaches with riboflavin: A systematic review. J Clin Pharm Ther, 2017), co-enzyme Q10 (please cite: Coenzyme Q10 supplementation for prophylaxis in adult patients with migraine-a meta-analysis. BMJ Open, 2021) and homocysteine lowering vitamins (please cite: Pyridoxine, folate and cobalamin for migraine: A systematic review. Acta Neurol Scand, 2020, AND Serum Homocysteine, Pyridoxine, Folate, and Vitamin B12 Levels in Migraine: Systematic Review and Meta-Analysis. Headache, 2020).

Design

Please report that the study was a ‘‘parallel group’’ RCT.

Participants

  1. Please move the sentence ‘‘19 persons were recruited in this study’’ to the Results section.
  2. Please provide the definition you used for migraine (ICHD-3?).
  3. The eligibility criteria should be described in detail. Please mention the age group of the participants (above 18 years?) Please state if chronic migraine, medication overuse headache and patients fulfilling criteria for more than one primary headache disorder were included. Also, please report if the use of prophylactic medication was considered in the eligibility criteria (and if any patients used preventive medication for migraine during the trial).

Procedure

Please describe the randomization process (table of random numbers? randomly generated sequence by a computer>). Also, please indicate if participants and/or investigators were blinded to the study procedures.

Table 1

Menstrual migraine is not regarded as a separate entity according to the ICHD-3 criteria. Please incorporate these patients to the corresponding group (probably the migraine without aura group). Moreover, please include gender information.

Exercise group

  1. Please report the settings of your study (health care center? Outside? Participants home?)
  2. Please describe strength exercises for upper and lower limbs, as well as TMJ exercises.

Control group

Did the control group follow the same program? The only difference was that the controls were unattended?

Statistical analysis

  1. Please combine the groups 0.0 < |r| ≤ 0.1 (insignificant) and 0.1 < |r| ≤ 0.39 (very weak) together and name the association as very weak. In the results you present multiple analyses.
  2. In case of multiple analyses we tend to use a corrected p-value = 0.05/(number of analyses). I suggest that you remove the within group analyses (their value is little) and keep only the between group analyses. Of course, you may keep the baseline and after-intervention values for each group.
  3. Also, among the between group comparisons, please indicate which outcomes are the primary (most important) and which are the secondary.

Results

  1. ‘‘The study showed a statistically significant cervical flexion results in exercise (p=0,007) and control (p=0,026) groups. Also, cervical extension (p=0,011) and lateral flexion to right (p=0,011) and to left (p=0,020) in the exercise group.’’ These information are not exactly relevant to migraine. Please use a composite score (e.g., the sum of the above scores) to pool all these scores (you can name it composite cervical motion).
  2. Moreover, have you performed any measurements regarding headache frequency, analgesic consumption and headache severity? The effect of exercise on these migraine parameters is important.

Discussion

  1. Please reformulate ‘‘Migraine is the primary headache’’ as follows: ‘‘Migraine is the 2nd commonest primary headache disorder’’.
  2. ‘‘causes the greatest disability in individuals under 50 years of age’’: Please specify compared to which disorders (primary headaches? neurological disorders? all disorders?)
  3. ‘‘However, in one study cervical range of motion was significantly improved compared to control group and no differences were found in groups and between the groups in another study’’. Please indicate the effect of physical therapy on migraine (the range of motion is not exactly relevant to migraine). Generally, please try to follow this recommendation (provide information relevant to migraine).

Conclusions and recommendations

Please remove the 1st and 2nd paragraph (they do not report conclusions and recommendations, but they report once again your results)

Table 5. Associations between cervical range of motion, pain, temporomandibular joint range of motion and quality of life in migraineurs

  1. These are too many arbitrary analyses. You only correlate a subgroup of the measured variables, but you do not describe the rationale (why were only these variables chosen to be analysed). You should provide a prespecified plan in the statistical analysis section and stick to it. I recommend that you limit the number of analyses (as suggested above). All these analyses tend to confuse the reader, who may lose track of what is important. However, if you choose to present all these analyses, please be consistent by reporting all possible correlations (do not choose which ones you will present).
  2. Also, in view of the number of analyses, I suggest that you use the 0.001 significance thresholds (performing multiple comparisons raises the probability that an association will be significant by chance, so we limit this probability by using stricter p value cut-offs).

Author Response

(The authors gave the same response as above.)

Round 2

Reviewer 1 Report

Authors addressed the raised comments and made some changes to the manuscript. However, I still have many concerns.

1 - From the Authors’ response, the main aim of the study is still not clear to me. I understand that the trial addressed the efficacy of physical exercise. However, Authors also state that the real difference between the experimental and control intervention were TMJ exercises. I suggest clarifying this point. Which is the intervention the Authors really wanted to assess? 

2 - When asked about reporting migraine outcomes, Authors answered that patients did not take any acute drugs within  this 6 weeks period. However, the outcomes that are usually assessed are migraine/headache days together with acute medication consumption, assessed prospectively before and after the intervention, usually with a headache diary. Those outcomes should be reported in a study of patients with migraine. The paper is intended to be read by headache physicians who care about headache-related outcomes. If not reported, this is a serious flaw of the study.

3 - A significant improvement was noted in several outcomes in both the experimental and the control group. This might be a point to discuss.

4 - The study strengths and limitations should be put in a paragraph at the end of the Discussion.

5 - I am totally aware of the limitations that the CoViD-19 pandemic imposed over medical research. I agree that it was difficult to find even only 19 study participants. However, Authors should still better discuss in the manuscript that the study groups were small and probably underpowered to show significant results in some of the outcomes. This point should be clearly stated. I suggest using the terms “pilot” or “preliminary” to underline that the Authors’ work must be expanded in future studies.

6 - Most patients took preventive migraine drugs during the study period. To ensure that Authors captured the efficacy of the experimental treatment only, those drugs must have been taken without any modification during the entire study period. Please clarify.

7 - Authors conclude that they “recommend” prescribing physical exercise to patients with migraine. However, the Authors’ suggestion cannot be considered a  “recommendation”, as this is a small study that cannot generate any strong recommendation.

Author Response

Please see comments in green colour in article

1 - From the Authors’ response, the main aim of the study is still not clear to me. I understand that the trial addressed the efficacy of physical exercise. However, Authors also state that the real difference between the experimental and control intervention were TMJ exercises. I suggest clarifying this point. What is the intervention the Authors really wanted to assess?  IMPROVED

2 - When asked about reporting migraine outcomes, Authors answered that patients did not take any acute drugs within this 6 weeks period. However, the outcomes that are usually assessed are migraine/headache days, together with acute medication consumption, assessed prospectively before and after the intervention, usually with a headache diary. Those outcomes should be reported in a study of patients with the migraine. The paper is intend to be read by neurologists who take care about headache-related outcomes. If not reported, this is a serious flaw of the study.

Unfortunately, mentioned migraine outcomes were unreported. Marked in study limitations.

3 - A significant improvement was noted in several outcomes in both the experimental and the control groups. This might be a point to discuss. Improvement in groups are discussed in Discussion paragraph.

4 - The study strengths and limitations should be put in a paragraph at the end of the Discussion. IMPROVED

 5 - I am totally aware of the limitations that the CoViD-19 pandemic imposed over medical research. I agree that it was difficult to find even only 19 study participants. However, Authors should still better discuss in the manuscript that the study groups were small and probably underpowered to show significant results in some of the outcomes. This point should be clearly stated. I suggest using the terms “pilot” or “preliminary” to underline that the Author‘s work must be expanded in future studies.  Marked in study limitations.

6 - Most patients took preventive migraine drugs during the study period. To ensure that Authors captured the efficacy of the experimental treatment only, those drugs must have been taken without any modification during the entire study period. Please clarify. Preventive migraine drugs such as vitamin supplements (vitamin B2, B6, B12)  were taken only.

7 - Authors conclude that they “recommend” prescribing physical exercise to patients with migraine. However, the Author‘s suggestion cannot be considered a “recommendation”, as this is a small study that cannot generate any strong recommendation. IMPROVED

Reviewer 2 Report

The revised manuscript is much improved compared to the initial version. However, additional English editing is required, as well as the following revisions.

Abstract

Please remove all references from the abstract.

Please explain all abbreviations in the abstract.

Moreover, results clearly indicate that the intervention benefitted pain and motion parameters of neck, upper back and temporomandibular muscles and joints. Regarding migraine parameters (headache severity, duration, frequency, etc) no information was provided. The only migraine-relevant parameter that you measured is the quality of life. Therefore, please modify your conclusion that ‘physical therapy is an effective tool in migraine treatment’ and recommendation to ‘include physiotherapy based on aerobic exercise in the treatment of migraine as an alternative and non-invasive method’. Based on your results, you could suggest to further test supervised physiotherapy in migraine parameters (because you cannot conclude that the improvement of other musculoskeletal parameters will benefit migraine itself). Also, please not that supervised physiotherapy is compared with unsupervised physiotherapy. As a result you should not recommend physiotherapy in general, since the unsupervised group may not benefit from physiotherapy.

Introduction

Please correct the following

‘one of the most common nervous system disorders causing the greatest disability in’

‘15-18 percent of the adult population is afflicted with migraine

‘are more likely to experience mood changes.’ (please remove than anxiety)

Please add the correct reference for vitamin B2. Also, using reference 10 please mention the potential effectiveness of vitamin D. Please note, that Vitamin D may benefit musculoskeletal parameters, similar to physiotherapy.

Discussion

Aerobic exercise benefits the musculoskeletal system, but may provide additional benefits for patients with migraine. Please enumerate some of these benefits: 1. Lipid-lowering (serum lipid abnormalities have been associated with migraine), please refer to [Serum lipid abnormalities in migraine: A meta-analysis of observational studies, Headache, 2021], 2. Reduction of oxidative and metabolic stress (oxidative and metabolic stressors such as homocysteine have been associated with migraine), please refer to [Serum Homocysteine, Pyridoxine, Folate, and Vitamin B12 Levels in Migraine: Systematic Review and Meta-Analysis, Headache, 2020], 3. Reduction of systemic inflammation (neuroinflammation plays a role in migraine), please refer to [Evaluation of Inflammatory State in Migraineurs: A Case-control Study, Iran J Allergy Asthma Immunol, 2020]

Conclusions and recommendations

Again, please modify your conclusions, as described in the Abstract.

Author Response

Thank you very much for your help and opinion. 

2nd reviewer

Please see comments in blue colour in article

Comments and Suggestions for Authors The revised manuscript is much improved compared to the initial version. However, additional English editing is required, as well as the following revisions. IMPROVED

Abstract Please remove all the references from the abstract. Please explain all abbreviations in the abstract. IMPROVED.

-Moreover, results clearly indicate that the intervention benefitted pain and motion parameters of neck, upper back and temporomandibular muscles and joints. Regarding migraine parameters (headache severity, duration, frequency, etc) no information was provided. The only migraine-relevant parameter that you measured is the quality of life. Mentioned outcomes (headache severity, duration, frequency) were not reported, marked in study limitations in green colour.

-Therefore, please modify your conclusion that ‘physical therapy is an effective tool in migraine treatment’ and recommendation to ‘include physiotherapy based on aerobic exercise in the treatment of migraine as an alternative and non-invasive method’.  IMPROVED, see green colour.

-Based on your results, you could suggest to further test supervised physiotherapy in migraine parameters (because you cannot conclude that the improvement of other musculoskeletal parameters will benefit migraine itself). IMPROVED

-Also, please not that supervised physiotherapy is compared with unsupervised physiotherapy. As a result you should not recommend physiotherapy in general, since the unsupervised group may not benefit from physiotherapy. IMPROVED

Introduction Please correct the following ‘one of the most common nervous system disorders causing the greatest disability in’ ‘15-18 percent of the adult population is afflicted with migraine’ ‘are more likely to experience mood changes.’ (please remove than anxiety) IMPROVED

-Please add the correct reference for vitamin B2. Also, using reference 10 please mention the potential effectiveness of vitamin D. Please note, that Vitamin D may benefit musculoskeletal parameters, similar to physiotherapy. IMPROVED

Discussion Aerobic exercise benefits the musculoskeletal system, but may provide additional benefits for patients with migraine. Please enumerate some of these benefits: 1. Lipid-lowering (serum lipid abnormalities have been associated with migraine), please refer to [Serum lipid abnormalities in migraine: A meta-analysis of observational studies, Headache, 2021], 2. Reduction of oxidative and metabolic stress (oxidative and metabolic stressors such as homocysteine have been associated with migraine), please refer to [Serum Homocysteine, Pyridoxine, Folate, and Vitamin B12 Levels in Migraine: Systematic Review and Meta-Analysis, Headache, 2020], 3. Reduction of systemic inflammation (neuroinflammation plays a role in migraine), please refer to [Evaluation of Inflammatory State in Migraineurs: A Case-control Study, Iran J Allergy Asthma Immunol, 2020] IMPROVED

Conclusions and recommendations Again, please modify your conclusions, as described in the Abstract. IMPROVED

Round 3

Reviewer 1 Report

Authors addressed all the raised comments. The study has indeed improved; however, some points remain to be improved.

1) Authors did not consider some of the main outcomes of migraine evaluation, such as migraine days and acute medication intake. The improvement in quality of life measures indirectly suggests a benefit of physical therapy in patients with migraine. However, this has to be confirmed in further studies. The present study is nothing more than a pilot study. This point should be clarified since the title.

2) It is interesting to note that the effect of physical therapy was significant even in the control group. This significance encourages further investigation in the field.

Author Response

Dear Reviewer, 

Thank you very much for your help and opinion.

Please see comments in green colour in article

1) Authors did not consider some of the main outcomes of migraine evaluation, such as migraine days and acute medication intake. The improvement in quality of life measures indirectly suggests a benefit of physical therapy in patients with migraine. However, this has to be confirmed in further studies. The present study is nothing more than a pilot study. This point should be clarified since the title. IMPROVED

2) It is interesting to note that the effect of physical therapy was significant even in the control group. This significance encourages further investigation in the field. IMPROVED